# Rabbit Monoclonal Antibody Specifically Recognizing a Linear Epitope in the RBD of SARS-CoV-2 Spike Protein

**DOI:** 10.3390/vaccines9080829

**Published:** 2021-07-28

**Authors:** Junping Hong, Qian Wang, Qian Wu, Junyu Chen, Xijing Wang, Yingbin Wang, Yixin Chen, Ningshao Xia

**Affiliations:** 1National Institute of Diagnostics and Vaccine Development in Infectious Diseases, School of Life Sciences, Xiamen University, Xiamen 361102, China; 21620180155490@stu.xmu.edu.cn (J.H.); 21620191152709@stu.xmu.edu.cn (Q.W.); wuqiann@stu.xmu.edu.cn (Q.W.); 21620200156503@stu.xmu.edu.cn (X.W.); nsxia@xmu.edu.cn (N.X.); 2State Key Laboratory of Molecular Vaccinology and Molecular Diagnostics, School of Public Health, Xiamen University, Xiamen 361102, China; JunyuChen@xmu.edu.cn

**Keywords:** SARS-CoV-2, spike protein, RBD, rabbit monoclonal antibody, IHC, linear B-cell epitope

## Abstract

To date, SARS-CoV-2 pandemic has caused more than 188 million infections and 4.06 million deaths worldwide. The receptor-binding domain (RBD) of the SARS-CoV-2 spike protein has been regarded as an important target for vaccine and therapeutics development because it plays a key role in binding the human cell receptor ACE2 that is required for viral entry. However, it is not easy to detect RBD in Western blot using polyclonal antibody, suggesting that RBD may form a complicated conformation under native condition and bear rare linear epitope. So far, no linear epitope on RBD is reported. Thus, a monoclonal antibody (mAb) that recognizes linear epitope on RBD will become valuable. In the present study, an RBD-specific rabbit antibody named 9E1 was isolated from peripheral blood mononuclear cells (PBMC) of immunized rabbit by RBD-specific single B cell sorting and mapped to a highly conserved linear epitope within twelve amino acids ^480^CNGVEGFNCYFP^491^ on RBD. 9E1 works well in Western blot on S protein and immunohistochemistry on the SARS-CoV-2 infected tissue sections. The results demonstrated that 9E1 can be used as a useful tool for pathological and functional studies of SARS-CoV-2.

## 1. Introduction

Coronavirus Disease 2019 (COVID-19) is a novel acute respiratory disease, caused by severe acute respiratory syndrome coronavirus 2 (SARS-CoV-2) [1,2]. The latest, SARS-CoV-2 has spread over 223 countries and resulted in more than 188 million infection cases and 4.06 million confirmed deaths worldwide (https://www.who.int/emergencies/diseases/novel-coronavirus-2019 accessed on 16 July 2021). There is an unmet medical need to develop more effective prophylactic and therapeutic approaches against SARS-CoV-2. Better detection reagents and methods including SARS-CoV-2-specific antibodies are also in urgent need to study of SARS-CoV-2 transmission, pathogenesis, and immune interventions.

SARS-CoV-2 is a positive-sense single-stranded RNA virus which belongs to lineage B β-coronaviruses. Like other coronavirus, SARS-CoV-2 encodes four structural proteins including spike (S) protein, envelope (E) protein, membrane (M) protein, and nucleocapsid (N) protein. The amino acid sequence homology between severe acute respiratory syndrome coronavirus-related coronavirus (SARS-CoV) and SASR-CoV-2 approximates to 76% for the spike (S) proteins [3]. S protein is the core fusion protein which is responsible for attaching the virus to cell surface and initiating virus entry [4,5]. SARS-CoV-2 and SARS-CoV share a common host-cell receptor protein, angiotensin converting enzyme 2 (ACE2), interacted by S protein for entry into the target cell [6,7,8]. S protein is divided into two subunits, S1 and S2, according to their functional differences. The receptor-binding domain (RBD) on S1 subunit plays a key role in binding to receptor on the surface of host cell and becomes an important target for functional study and vaccine and therapeutics development [9]. However, it is not easy to detect RBD in Western blot using polyclonal antibody, suggesting that RBD may form a complicated conformation under native condition and have rare linear epitope. Thus, it is essential to develop a monoclonal antibody (mAb) that recognizes linear epitope on RBD, which will be useful for the RBD detection.

In this study, we have isolated a SARS-CoV-2 RBD -specific rabbit antibody 9E1, recognizing a linear B-cell epitope, which can be used in ELISA, Western Blot, and immunohistochemistry (IHC) assay. In fact, there is an urgent need to develop a good IHC antibody for the study of pathology of COVID-19 patients and SARS-CoV-2-infected animal model [10,11,12]. 9E1 could be a useful tool for pathological and functional studies of SARS-CoV-2.

## 2. Materials and Methods

### 2.1. Clone and Expression of SARS-CoV-2 RBD and S Protein

The gene of SARS-CoV-2 RBD protein and S protein with two stabilizing proline mutations at residues 986, 987 and a “GSAS” substitution at the furin cleavage site (GenBank accession no. MN908947.3) were synthesized and cloned into pCDNA3.1 expression vector using Gibson assembly [4]. Plasmids were transfected into 293F cells with Polyethyleneimine (PEI). Proteins were expressed for 5 days and supernatant was collected for further purification using Ni-NTA affinity chromatography column (GE Healthcare, Boston, MA, USA).

### 2.2. Rabbit Immunization

10-week-old New Zealand white rabbits were obtained from Songlian Laboratory Animal Center, Songjiang District, Shanghai. The immunogen (300 μg RBD protein) dissolved in 1 mL PBS were emulsified in an equal volume of complete Freund’s adjuvant (Sigma, St. Louis, MO, USA). Rabbits were injected subcutaneously by multiple sites. Two and four weeks after the first immunization, the boosts were conducted by immunization with 300 μg RBD protein mixed with incomplete Freund’s complete adjuvant (Sigma-Aldrich, St. Louis, MO, USA).

### 2.3. Antibody Screening

After immunization, approximately 5 mL blood was collected from immunized rabbits. Peripheral blood mononuclear cells (PBMCs) were isolated by Ficoll-Paque PLUS (GE Healthcare, Boston, MA, USA) according to the protocol provided by the manufacturer. Briefly, the whole blood of rabbit was diluted with serum-free RPMI 1640 at a ratio of 1:1. The mixture was layered onto Ficoll-Paque PLUS (GE Healthcare, Boston, MA, USA) in 15 mL sterile tubes and centrifuged at 800× *g* for 30 min at room temperature. The PBMCs layer was collected and washed three times in PBS. We used Sulfo-NHS-LC-Biotin (Thermo Fisher Scientific, Waltham, MA, USA) conjugated RBD protein to sort antigen-specific B cells. PBMCs were suspended in 100µL PBS and incubated with biotin conjugated RBD for 30 min at 4 °C. The mixture was then washed in PBS and pelleted by centrifugation at 800× *g* for 5 min

PBMCs were labeled by a panel of reagents: LIVE/DEAD Aqua (Thermo Fisher Scientific, Waltham, MA, USA), Mouse anti Rabbit CD4:FITC (Bio-Rad, Hercules, CA, USA), Mouse anti Rabbit CD8:FITC (Bio-Rad, Hercules, CA, USA), Mouse anti Rabbit T Lymphocytes:FITC (Bio-Rad, Hercules, USA),Mouse anti Rabbit IgM:RPE (Bio-Rad, Hercules, CA, USA), Streptavidin APC Conjugate (Thermo Fisher Scientific, Waltham, MA, USA).

B cell sorting was carried out on a BD Biosciences FACS Aria III. Lymphocytes were gated by size and granularity using FSC vs. SSC. Single lymphocytes were chosen using FSC-H vs. FSC-A. Dead cells and T-cells (FITC) were identified and excluded. B cells were identified with PE anti-rabbit IgM antibody and BV421 anti-rabbit IgG antibody staining. Antigen specific B cells were labeled with APC conjugated RBD protein. Target cells were sorted into a 96-well plate containing lysis buffer.

Positive B cells were lysed and RNA was extracted. cDNA was prepared using Superscript III reverse transcriptase (Invitrogen, Carlsbad, CA, USA) as described previously [13]. Antibody variable region genes were then recovered via two rounds of PCR using GXL polymerase (TaKaRa, Dalian, China) and then inserted into VRC8400 vectors containing the heavy chain and light chain constant region of the rabbit IgG subtype antibody. The recombinant antibodies were expressed in 293F cells through transient transfection and purified from culture media through metal affinity chromatography using Ni-NTA resin (GE Healthcare, Boston, MA, USA).

### 2.4. ELISA

100 ng/well of proteins in 0.1 M carbonate buffer (pH 9.6) was coated on 96-well microplates separately at 4 °C overnight and blocked with 2% skim milk for 2 h at 37 °C. After washing three times with PBS containing 0.5% Tween-20 (PBST), 100 μL of serially diluted antibody or supernatant was added to the wells and incubated at 37 °C for 30 min. After five washes, 100 μL of horseradish peroxidase (HRP)-conjugated goat anti-rabbit IgG secondary antibody solution was added to each well and incubated at 37 °C for 30 min. After five washes, 100 μL of tetramethylbenzidine (TMB) substrate (Wantai BioPharm, Beijing, China) was added and incubated at 37 °C for 15 min in the dark. Then, the reaction was stopped with 50 μL 2M H_2_SO_4_ solution and the absorbance was measured at 450 nm.

Competition ELISA was carried out with an additional step involving preincubation of synthesized peptides (1 μg) with HRP conjugate mAb 9E1 at 37 °C for 2 h. The mixture was added to the RBD coated plates and incubated at 37 °C for 30 min. After washing, TMB substrates were added and stopped with 2M H_2_SO_4_. The OD value was determined at 450 nm. If the peptides had reactivity with mAb 9E1, the OD value would be low. If the peptides had no reactivity with mAb 9E1, the OD value would be high.

### 2.5. Western Blot

Purified RBD and S proteins were subjected to 12% SDS-PAGE gels and then transferred to 0.22μm nitrocellulose (GE Healthcare, Boston, MA, USA). The blotted membranes were blocked with 5% skim milk in tris-buffered saline and then incubated with mAb 9E1 at 37 °C for 60 min. After washing three times with PBST, the membranes were incubated with an HRP-conjugated goat anti-mouse antibody at 37 °C for 60 min and visualized using chemiluminescent HRP substrates (Thermo Fisher Scientific, Waltham, MA, USA).

### 2.6. Prokaryotic Expression of Recombinant Truncated Rbd Protein

For the expression of recombinant truncated RBD proteins, the truncated RBD gene was cloned into pTO-T7 vector and transferred into ER2566 *E. coli* strain (Invitrogen) for further expression. The construct contains an N-terminal CRMA fusion protein through a flexible peptide linker ((G_4_S)_3_) to improve the antigenicity of protein and a C-terminal 6-His tag for purification. The bacteria were centrifuged and lysed by ultrasonication. The CRMA- (truncated RBD) protein was purified from de-natured inclusion bodies by dialyzing into neutral buffer for renaturation as described previously [14].

### 2.7. Identification of SARS-CoV-2 Positive Tissues

To identify the positivity of SARS-CoV-2 infected tissues, hamster tissue samples were collected from our previous animal study of SARS-CoV-2 vaccine development and homogenized by a TissueLyser II (Qiagen, Hilden, Germany). Viral RNA was extracted using a QIAamp Viral RNA Mini Kit (Qiagen, Hilden, Germany) according to the manufacturer’s instructions. Viral RNA quantification was conducted using SARS-CoV-2 RT-PCR Kit (Wantai BioPharm, Beijing, China).

### 2.8. Immunohistochemistry

The sections of lung tissues from SARS-CoV-2 infected hamsters were collected from our previous animal study of SARS-CoV-2 vaccine development. The expression of S in lung histological sections was examined by immunohistochemical staining. Sections were blocked with 1% bovine serum albumin in PBS, stained with mAb 9E1 at a dilution of 1:3000 overnight at 4 °C and then incubated with goat anti-rabbit IgG H- and L chain-specific biotin conjugate (Calbiochem, Darmstadt, Germany) at a dilution of 1:2000 for 30 min at room temperature. Tissue sections were then incubated with streptavidin/peroxidase complex reagent (Vector Laboratories, CA, USA) for 30 min at room temperature, and color was developed using 3,30-diaminobenzidine (Vector Laboratories, Burlingame, California, USA) according to the manufacturer’s instructions.

### 2.9. Epitope Alignment

To assess the degree of homology of the epitope recognized by mAb9 9E1, more than 1000 strains of SARS-CoV-2 were downloaded from GISAID (https://www.gisaid.org/ accessed on 1 May 2020). The amino acid sequences of S protein were aligned and analyzed using MEGA software.

### 2.10. Clinical Samples and Ethical Statement

Three patients’ plasma was collected from COVID-19 patients after they recovered from the disease in the First Affiliated Hospital of Xiamen University. The first patient was a woman aged 74. The second patient was a woman aged 33 years. The third patient was a man aged 44 years. This study was approved by the medical ethics committee at Xiamen University, China (#2020-11-12) and a written informed consent was obtained from each patient.

## 3. Results

### 3.1. Isolation of SARS-CoV-2 S Protein Specific Rabbit Antibody

Spike (S) protein is the core fusion protein for SARS-CoV-2 to infect host cell. Receptor binding domain (RBD) on S is essential for host cell receptor ACE2 binding. We construct two clones based on pCDNA3.1 vector to express RBD (aa316–549) and S protein with two stabilizing proline mutation in the C-terminal S2 fusion machinery according to previous report using 293F cells (Figure 1A,B) [4]. The apparent molecular mass of purified RBD and S protein is ~37 KDa and ~180 KDa, respectively in SDS-PAGE under reducing conditions (Figure 1C,E). Anti-His tag specific antibody reacted to corresponding bands in Western blot (WB) assay (Figure 1C,E). Both RBD and S protein were recognized by convalescent COVID-19 human sera, indicating that they were properly folded and exhibited native antigenicity (Figure 1D,F and Appendix A).

SARS-CoV-2 RBD protein was used as sequential immunogens for rabbit immunization. Peripheral blood mononuclear cells (PBMCs) were isolated from whole rabbit blood and then stained using the antibody panel described in Table 1. The panel included the use of Aqua to remove dead cells, an anti-T cells antibody pool to eliminate T cells. We employed negative stain for IgM to exclude naïve B cells. We included an anti-IgG antibody to ensure staining of memory B cells expressing IgG. We used APC-labeled SARS-CoV-2 RBD as probe bait to screen and isolate RBD-specific B cell. The gating strategy for sorting antigen specific B cells was displayed in Figure 2A. RBD-specific B cells were rare in IgG positive B cells which was only 0.10% (Figure 2A). The sorted B cells were collected and lysed for reverse transcription, two rounds of PCR amplification and further variable chain genes sequencing. Paired sequences were cloned into expression vectors with constant regions of rabbit IgG subtype heavy and kappa chain, respectively for further expression and evaluation. We generated a panel of eight RBD-specific rabbit antibodies (Figure 2B,C).

### 3.2. Reactivity of 9E1 with RBD and S Protein

To know the reactivity of RBD-specific rabbit antibodies, we characterized these antibodies by using Western blot. Western blot showed that only 9E1 could react with denature RBD protein among these antibodies (Figure 3A), indicating that 9E1 bound to a linear epitope on RBD protein. 9E1 had a good reactivity with RBD and S protein in both Western blot and ELISA assay (Figure 3B–D). To verify the specificity of 9E1 against the SARS-CoV-2 viruses, formalin-fixed paraffin-embedded (FFPE) lung tissue of virus infected-hamster was stained by 9E1 and control antibody in immunohistochemistry assay (IHC). In the lungs, 9E1 was able to detect the viral S proteins expressed in alveolar epithelia cells while control antibody failed (Figure 3E). This result indicated that S specific mAb 9E1 can be used as a specific immunohistochemistry tool to study the distribution and function of S protein in the infection of SARS-CoV-2 viruses.

### 3.3. Epitope Mapping of 9E1

To identify the epitope of 9E1, six truncated fragments covering the full-length RBD with overlapping regions were expressed as CRMA-fused proteins using the prokaryotic expression system. As showed in the Western blot analysis, 9E1 could recognize three truncated fragments (S-406-505aa, S-436-535aa, S-466-549aa), suggesting the epitope was included within 466–505aa (Figure 4A and Appendix A). To further identify the epitope, the amino acid residues between 466aa and 505aa were divided into three sections (S-466-485aa, S-476-495aa, S-486-505aa). The three sections were 20 amino acids residues in length with 10 amino acid residues overlapping. Competition ELISA revealed that the peptide S-476-495aa inhibited the binding of 9E1 to RBD, indicating 476–495aa as the epitope (Figure 4B and Appendix A). To minimize the location of 9E1 epitope further, a panel of short peptides was synthesized for competition ELISA, and the result showed that sequence 480–491aa could be the precise epitope (Figure 4C and Appendix A). To confirm the epitope, another panel of short peptides (P1–P13) was synthesized, further competition ELISA results showed that the core sequence recognized by 9E1 was P2 (^480^CNGVEGFNCYFP^491^) (Figure 4D and Appendix A). The deletion of ^480^C or ^491^P of P2 would abrogate the binding activity of the peptides with 9E1. To investigate the conservation of 9E1 epitope among different SARS-CoV-2 strains, more than 1000 strains deposited in GISAID (https://www.gisaid.org/ accessed on 1 May 2020) during the early stage of the pandemic were aligned and analyzed. We found out that glycoprotein E and M are highly conserved. Several mutations are found in S protein sequences in part of the isolates. The align result showed that the 9E1 recognizing epitope was highly conserved among selected strains (Figure 4E).

9E1 recognizing linear B-cell epitope has been defined between 480aa and 491aa of S protein. The sequence was mapped in the 3D model based on X-ray crystal structure of S protein of SARS-CoV-2 in prefusion state (PDB ID: 6z97). The image showed that 9E1 binding site was located in the surface of S protein and included in the receptor binding motif (RBM), suggesting the region in S protein is accessible under native state.

## 4. Discussion

The COVID-19 pandemic is spreading rapidly around the world and becoming a great global health burden. Suitable detection reagents and methods including SARS-CoV-2-specific antibodies are in urgent need to study of SARS-CoV-2 transmission, pathogenesis and immune interventions. A recent study had collected six commercially available SARS-CoV-associated antibodies and one monoclonal antibody against SARS-CoV-2 S to test whether they could be used in SARS-CoV-2 related immune detection; however, there were only two antibodies could specifically stain SARS-CoV-2 positive formalin-fixed and paraffin-embedded (FFPE) cell pellets in IHC and IFA assay, including a rabbit polyclonal antibody against S protein of SARS-CoV and a mouse monoclonal antibody against NP protein of SARS-CoV [15]. Polyclonal antibody is a mixture of monoclonal antibodies which may suffer from batch-to-batch variability and has potential cross-reactivity for recognizing multiple epitopes. Hence, it is valuable to develop highly specific SARS-CoV-2 antibodies for viral detection, such as Western Blot and IHC.

Receptor binding domain (RBD) of SARS-CoV-2 Spike protein was immunodominant in rabbits [16]. Compared to other spike immunogens (S1, S2, and S1 + S2), RBD could elicit a higher antibody titer and higher affinity antibodies to spike protein. In this study, RBD protein derived from the SARS-CoV-2 virus were used for rabbit immunization to generate a panel of rabbit monoclonal antibodies. One of the mAbs, 9E1, had a good reactivity to denatured RBD and S protein in Western blot assay and was further mapped to a conserved linear B-cell epitope within twelve amino acids (^480^CNGVEGFNCYFP^491^ in RBD of SARS-CoV-2). This is the first linear epitope on RBD identified so far.

Recently, the linear B cell epitopes on S protein were reported based on antibody profiles of COVID-19 patients [17,18,19]. The depletion of antibodies to two linear epitopes significantly reduced neutralization capacities indicating these two regions as critical neutralizing B-cell epitopes [17]. Several peptides displayed high specificity and sensitivity for detecting SARS-CoV-2 infection [18]. Moreover, some epitopes show strong association with disease severity and clinical outcome [18]. The linear 9E1 epitope on RBD identified here would be helpful for the understanding of the spike protein structure and function in the infection of SARS-CoV-2 virus. Based on the 3D structure model of spike protein, the 9E1 linear B-cell epitope is located in the surface of spike protein without coverage by other domains, showing this epitope is accessible in both up and down conformation of RBD. The sequence analysis also suggested that amino acid sequence of the 9E1 epitope was highly conserved in more than 1000 strains of SARS-CoV-2, while the epitope in SARS-CoV-2 differed from corresponding region of S protein from SARS-CoV (Appendix A), which indicating 9E1 is a SARS-CoV-2 specific antibody and have no cross-reactivity to SARS-CoV and other human coronaviruses S protein. It will be interesting to explore whether the corresponding region in SARS-CoV S protein could generate 9E1-like antibodies which may also recognize a linear epitope. Since this antigenic epitope has not been reported before, additional investigation is needed to determine whether this epitope has function in viral entry and replication and whether the epitope is a new antiviral target for drug design.

Additionally, this epitope might be suitable to develop diagnostic kit due to its high degree of homology among SARS-CoV-2 isolates. IHC results could especially demonstrate the infection process of viruses and advance our understanding of pathology of diseases like COVID-19 [20,21,22]. However, there is few RBD-specific antibodies applied in IHC assay [15,23]. In our study, the IHC result showed that 9E1 could stain the cells infected by SARS-CoV-2 suggesting that 9E1 could be used as a specific immunochemical tool to study the function of S in the replication of SARS-CoV-2. Further evaluation of 9E1 reactivity in human tissue is needed to explore its clinical value of detecting SARS-CoV-2.

In summary, we had generated a rabbit monoclonal antibody, 9E1, binding to RBD on S protein of SARS-CoV-2. 9E1 had good reactivity in ELISA, Western blot, and IHC assay. The novel linear B-cell epitope recognized by 9E1 was identified as 480–491 amino acid residues of S protein. The present study showed that 9E1 could be a useful tool for studying SARS-CoV-2 S protein functions as well as some clinical testing applications.

## Figures and Tables

**Figure 1 vaccines-09-00829-f001:**
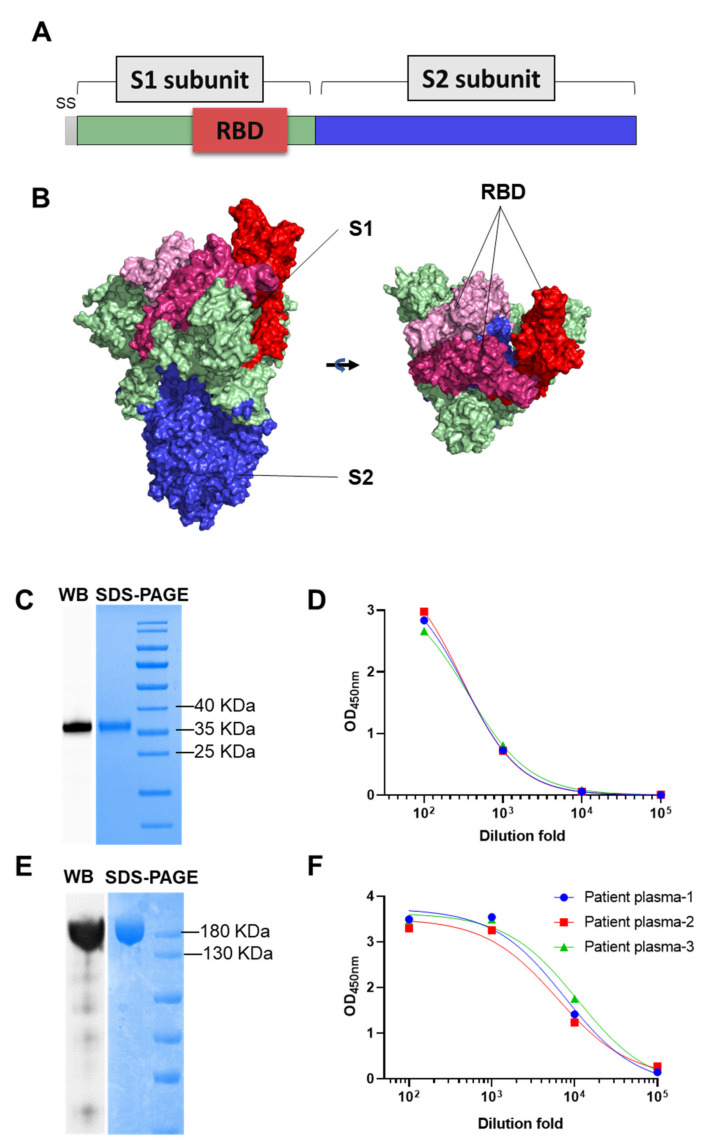
Biochemical and antigenic characterization of purified RBD and S protein. (**A**,**B**) Graphical representation of S protein (S1 and S2) and receptor binding domain (RBD). S1 subunit in light green, S2 in blue, RBD in red. Images were edited with PyMOL and the PDB accession number of S protein is 6z97. (**C**,**D**) SDS-PAGE, Western blot (WB) of purified RBD and RBD ELISA reactivity with COVID-19 convalescent sera. (**E**,**F**) SDS-PAGE, Western blot (WB) of purified S protein and SELISA reactivity with COVID-19 convalescent sera. Anti-His antibody was used as detection antibody in Western blot.

**Figure 2 vaccines-09-00829-f002:**
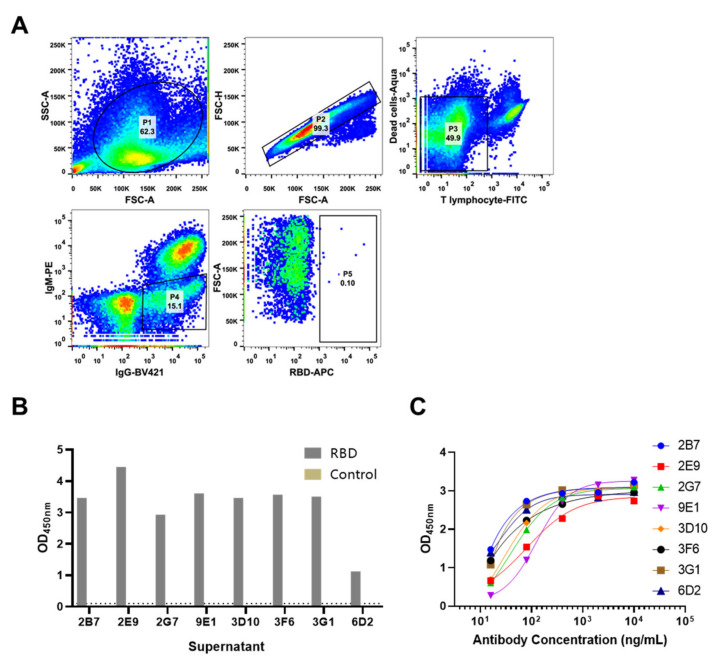
Isolation of RBD-specific rabbit antibodies by FACS. (**A**)PBMCs separated from RBD-immunized rabbit whole blood were stained with RBD conjugated to streptavidin-APC, viability dye, and antibodies specific for CD4, CD8, T lymphocyte, IgM, IgG. (**B**)The transfected supernatant of isolated rabbit monoclonal antibodies was evaluated for reactivity against RBD protein in ELISA assay. (**C**) Purified rabbit monoclonal antibodies were evaluated in ELISA assay with serial dilution.

**Figure 3 vaccines-09-00829-f003:**
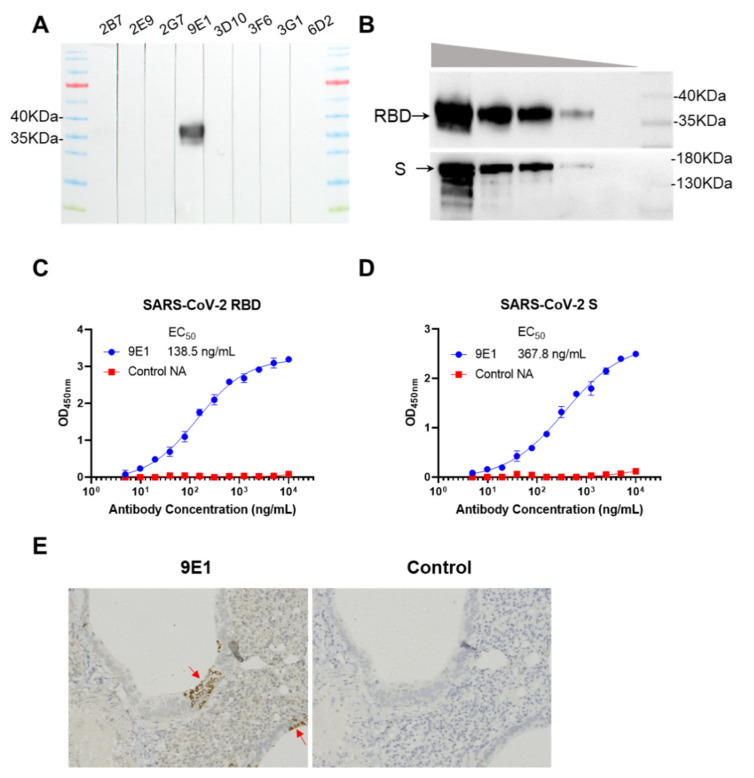
Reactivity of 9E1 with purified RBD and S protein. (**A**) Western blot reactivity of isolated rabbit antibodies with denatured RBD protein. (**B**) 9E1 was tested by Western blot for sensitivity against RBD (up) and S (down) proteins. From lane left to lane right, 2, 1, 0.5, 0.1, 0.01μg protein was loaded separately. (**C**,**D**) Purified 9E1 were diluted from 10 μg/mL to 5 ng/mL to measure the reactivity with RBD and S protein in ELISA assay. (**E**) The detection of SARS-CoV-2 in the lung tissue of infected hamster. Hamster was infected by SARS-CoV-2 with 10^4 PFU and euthanized 3 days later. Lung of the hamster was collected and tested by immunohistochemistry. 9E1 was used as primary antibody with dilution rate 1:3000.

**Figure 4 vaccines-09-00829-f004:**
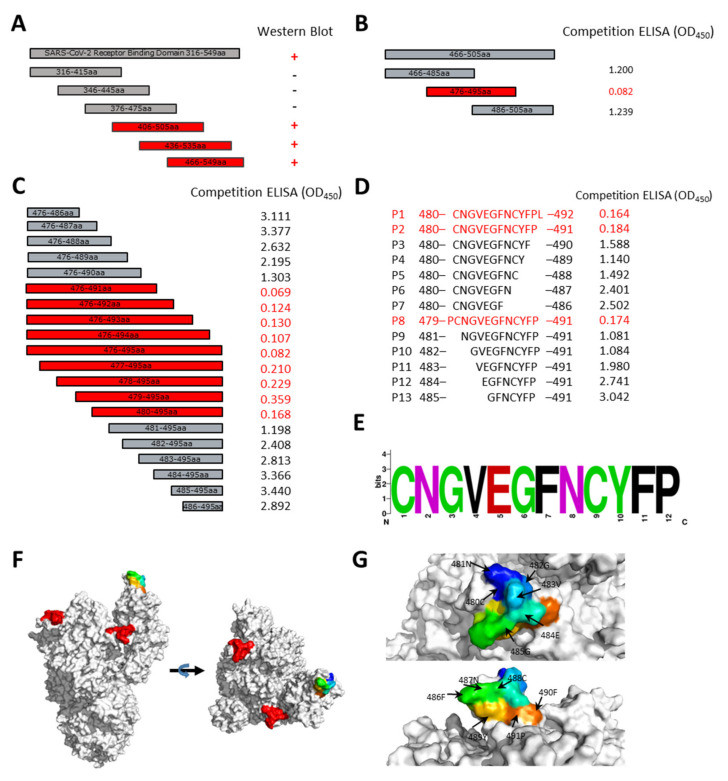
Epitope identification of 9E1. (**A**) Western blot result of 9E1 with overlapping RBD fragments. Color in red (+) means positive reactivity, color in gray (-) means negative reactivity. (**B**) Competition ELISA of 9E1 with three truncated peptides. (**C**) Competition ELISA of 9E1 with a panel of synthesized peptides. (**D**) Reconfirmation of 9E1 recognizing epitope with competition ELISA. 9E1 was mixed with peptides, and then the mixture was tested in indirect ELISA. The peptide which can block the binding of 9E1 to RBD was showed in red (low OD value). (**E**) Conservation analysis of 9E1 recognized epitope (**F**). The epitope of 9E1 was illustrated on the 3D structure of S Trimer in different angles (labeled with red and rainbow colors). (**G**) The detail of epitope recognized by 9E1 is displayed. Amino acids were labeled in variable colors. Images were edited with PyMOL and the PDB accession number of S protein is 6z97.

**Table 1 vaccines-09-00829-t001:** Summary of reagents used for isolation of IgG+ antigen specific rabbit B cells.

Marker/Reagent	Cell Type	Fluorophore
Aqua	Dead cells	
CD4	T cells	FITC
CD8	T cells	FITC
T Lymphocytes	T cells	FITC
IgM	Naïve B cells	RPE
IgG	Memory B cells	BV421
Antigen	Ag specific B cell	APC

A panel of fluorophore-labeled reagents was used for identification of dead cells, T cells, IgM positive B cells, IgG positive B cells, and antigen specific B cells.

## Data Availability

Not applicable.

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
