# Peer review of "Rabbit Monoclonal Antibody Specifically Recognizing a Linear Epitope in the RBD of SARS-CoV-2 Spike Protein"

_vaccines, 2021, doi:10.3390/vaccines9080829_

Round 1

Reviewer 1 Report

A rabbit antibody recognizing a conserved linear B-cell epitope on spike protein of SARS-CoV-2

Junping Hong , Qian Wang , Qian Wu , Junyu Chen , Xijing Wang , Yingbin Wang * , Yixin Chen * , Ningshao Xia

REVIEWER COMMENTS

This is a most interesting paper emerging out of an extensive systematic investigation on the "monospecificity" or "private epitope recognition" by a Rabbit monoclonal antibody (mAb). The mAb, designated as 9E1,  is a monospecific monoclonal antibody that recognizes amino acid sequence between 476 and 495 in the RBD domain of the S1 subunit of the SARS-CoV-2 protein that binds to Angiotensin convertase 2 (Receptor). The investigation was carried out elaborately.

Considering the strenuous efforts taken by the authors, it comes to mind, why not the authors take the patients B cells and immortalize them by several means, including EBV and develop human monoclonal antibodies directly and identify the monospecificity of the mAb, as has been done in this study. What is the need for such circumvented effort to raise the mAb in a rabbit?  However, this is just a suggestion for this outstanding investigation.  Most important several anti-covid antibodies are claimed in the literature, and one or two could be monospecific. The most important criterion of an antibody is whether it is MONOSPECIFIC or not.   In this respect, the authors systematically provided the critical pieces of evidence to prove the monospecificity of mAb 9E1.  The epitope mapping is well done.

All scientific papers will have some pitfalls. The quality of the work and the outcome of the results supersede the pitfalls; They are minor in the manuscript. I wish to bring out some such pitfalls in the manuscript.

  1. In my opinion, the paper will be complete if the authors could give a 3-dimentional drawing of the domain of S1 and S2 subunits, showing RBDs in the protein. Is there One RBD or several RBDs? Such figure will replace figure 1A!
  2. While describing Epitope mapping results in section 3.3, it would be beneficial to give a detailed RBD sequence to show that the specific epitope or their fragments are not found elsewhere in the RBD sequence.
  3. Every immunologist is aware that the epitope recognized by a B cell is linear, unlike that of T cell epitopes. Then why this talk of linearity even in the Title. To me, it reads terrible. The authors should focus on monospecifiicity of the mAb rather than the linearity.

Some aspects concerning Material and Methods are confusing.

  1. Page 3, line 98, the authors should clarify "Positive B cells" in the present context. The whole paragraph (lines 98 starting with Positive B cells and ending in n… Boston USA) should be simplified for a better understanding of the principle involved.  Clear presentation of how they have identified the Positive B cells would be beneficial to readers
  2. Section 2.6 Prokaryotic …..RBD protein. This section needs oversimplification, for it is a burden to the reader to make it out. For those experienced in this area of research, it is easy to read, but some of my fellow scientists felt that the language needs editing here.
  3. The amino acid sequence of S protein of 1100 strains can be aligned in an Excel file. They can visually evaluate for the MEGA software may not be available in all labs. The reviewer still wonders whether, after such a detailed screening, can't they confirm that the epitope is specific for SARS-C0V2-?
  4. The authors do not discuss diversification, if any, seen in the sequences of 1100 strains.  Do the strains they examined differences in their sequences?- this is a fundamental question. It is necessary to mention their findings in a sentence or two.  Otherwise, it casts doubt on their investigation regarding strains.
  5. We now know that the authors have elegantly mapped the sequence recognized by mAb 9$1. While this so, why not change the title to SARS-COV-2 RBD DOMAIN MONOSPECIFIC RABBIT MONOCLONAL IgG ANTIBODY. What is the subclass of mAb IgG1? IgG2a? IgG3?
  6. Figure 2A needs clarity and should be crisp and clear.

Introduction and Discussion are well written.

A MAJOR CONCERN:

  1. THE AUTHORS HAVE USED 3 PATIENTS' PLASMA. DID INSITUTIONAL IRB APPROVE THAT. WHAT IS THE NUMBER, IT IS IMPORTANT THE AUTHORS SHOULD MENTION THE AGE OF EACH PATIENT, FOR IMMUNOGENICITY MAY VARY WITH AGE. This information is essential for revision.

Author Response

Response to comments from Reviewer 1:

  1. In my opinion, the paper will be complete if the authors could give a 3-dimentional drawing of the domain of S1 and S2 subunits, showing RBDs in the protein. Is there One RBD or several RBDs? Such figure will replace figure 1A!

Response 1:

Thank you for the good suggestion. We have added a 3-dimentional model to display S1, S2 and RBD domains of SARS-CoV-2 in Figure 1, please see the revision in Page 5.

  1. While describing Epitope mapping results in section 3.3, it would be beneficial to give a detailed RBD sequence to show that the specific epitope or their fragments are not found elsewhere in the RBD sequence.

Response 2:

For the section 3.3, we used six truncated fragments and 20 peptides derived from RBD to map the 9E1 epitope (as shown in Figure 4 in Page 11). As suggested, we have summarized the detailed sequence of RBD, truncated fragments or peptides in Table S1 (please see the supplementary pdf file), showing that the unique epitope recognized by 9E1 is located between 480 and 491 amino acids in RBD and not found elsewhere.

  1. Every immunologist is aware that the epitope recognized by a B cell is linear, unlike that of T cell epitopes. Then why this talk of linearity even in the Title. To me, it reads terrible. The authors should focus on monospecifiicity of the mAb rather than the linearity.

Response 3:

Thank you for the suggestion. Now the revised title is “A monospecific rabbit antibody recognizing a unique epitope in the RBD of SARS-CoV-2 Spike protein”.

  1. Page 3, line 98, the authors should clarify "Positive B cells" in the present context. The whole paragraph (lines 98 starting with Positive B cells and ending in n… Boston USA) should be simplified for a better understanding of the principle involved.  Clear presentation of how they have identified the Positive B cells would be beneficial to readers

Response 4:

Thank you for the comments. We have re-written this part, please see the revision in Page 5. Lines 95-101.

  1. Section 2.6 Prokaryotic …..RBD protein. This section needs oversimplification, for it is a burden to the reader to make it out. For those experienced in this area of research, it is easy to read, but some of my fellow scientists felt that the language needs editing here.

Response 5:

Thank you for the comments, Section 2.6 has been revised in the text, please see the revision in Lines 135-141, Page 3.

  1. The amino acid sequence of S protein of 1100 strains can be aligned in an Excel file. They can visually evaluate for the MEGA software may not be available in all labs. The reviewer still wonders whether, after such a detailed screening, can't they confirm that the epitope is specific for SARS-C0V2-?

Response 6:

To make sure that the 9E1 epitope is specific to SARS-CoV-2, we aligned the S protein sequences of SARS-CoV-2, SARS-CoV, and five other human coronaviruses in Figure S5 (See it in supplementary PDF file), showing that 480CNGVEGFNCYFP491 (marked in red) in SARS-CoV-2 S protein sequence is different from the corresponding sequence of other coronaviruses S proteins. We also performed ELISA assay to confirm that 9E1 had no cross-reaction with S -RBD protein of SARS-CoV, please see the Figure S4 in supplementary PDF file. Hence, we conclude that the 9E1 epitope is specific to SARS-CoV-2.

  1. The authors do not discuss diversification, if any, seen in the sequences of 1100 strains.  Do the strains they examined differences in their sequences?- this is a fundamental question. It is necessary to mention their findings in a sentence or two.  Otherwise, it casts doubt on their investigation regarding strains.

Response 7:

Thank you for the comments. In fact, we found out that glycoprotein E and M are highly conserved and several mutations have been found in S protein sequences in some SARS-CoV-2 isolates. Please see the related revision in Page 11, Lines 266-269.

  1. We now know that the authors have elegantly mapped the sequence recognized by mAb 9E1. While this so, why not change the title to SARS-COV-2 RBD DOMAIN MONOSPECIFIC RABBIT MONOCLONAL IgG ANTIBODY. What is the subclass of mAb IgG1? IgG2a? IgG3?

Response 8:

Thank you for the suggestion. The title has been changed to “A monospecific rabbit antibody recognizing a unique epitope in the RBD of SARS-CoV-2 Spike protein”.

For the isotype of rabbit antibody, we explain that rabbits only have one IgG subclass, please refer to a published review below if necessary.

Weber J; Peng H; Rader C. From rabbit antibody repertoires to rabbit monoclonal antibodies. Exp Mol Med. 2017; 49(3):e305 (ISSN: 2092-6413).

  1. Figure 2A needs clarity and should be crisp and clear.

Response 9:

Thank you for the suggestion. Please see the new Figure 2A in Page 8.

  1. THE AUTHORS HAVE USED 3 PATIENTS' PLASMA. DID INSITUTIONAL IRB APPROVE THAT. WHAT IS THE NUMBER, IT IS IMPORTANT THE AUTHORS SHOULD MENTION THE AGE OF EACH PATIENT, FOR IMMUNOGENICITY MAY VARY WITH AGE. This information is essential for revision.

Response 10:

Thank you for the comments. The related information has been added in the text, please see the revision in Page 4, Line 168-174. 

Reviewer 2 Report

In this manuscript Hong et al., reported the isolation and characterization of a rabbit anti-RBD antibody, 9E1, that binds to the linear epitope that is conserved across the different variants of SARS-CoV-2. Using different peptides, the authors clearly identify the amino acid sequences that interact with the antibody. Using different binding assays, the authors also report that 9E1 recognizes the native and denatured form of the epitope in Spike protein and RBD. The manuscript is well written, bringing key questions that are addressed in the results part. I have a few comments that I would like to bring to attention to improve the quality of the manuscript.

  1. In Fig 1C and 1D, the authors show binding of polyclonal antibodies from patient plasma to RBD and Spike protein. Here the negative controls are missing. It is critical to perform the same assay using a plasma sample from non-exposed volunteer or ideally a plasma sample collected from pre-pandemic time point.
  2. In Fig2A, the FACS panels should be depicted not always as histograms but as dot plot. This is particularly relevant to appreciate the cell frequency and the MFI values in the IgG+ cells stained for RBD-APC.
  3. In Fig 2B, what was the concentration of antibodies tested for binding to RBD? This needs to be mentioned both in the methods and the figure legend. Also, were these experiments performed at serially diluted antibody concentrations as described in the method? If yes, could the author show the dilution curves of all antibodies? It is clear all of them bind to RBD and 9E1 may not be the best antibody. It would be good to compare the binding strength of the antibodies and then show how 9E1 is the only antibody that binds in Western blot.
  4. In Fig 3E, the positive control is missing in IHC.
  5. In Fig 4. Panel A shows the peptide sequence covered from the protein and whether 9E1 binds to the peptide in Western blot. Here I would recommend the authors to write the amino acid sequences of the peptides used (as shown in panel D) or at least provide them in supplementary figures. In addition, it is important to show the raw data of Western blot as this is a crucial data supporting the message of the manuscript. The authors should show the raw Western blot data of binding to the peptides in comparison to the whole protein.
  6. Similarly, in Fig 4B, the peptide amino acid sequence should be shown and the competition ELISA values should be shown in comparison to the corresponding controls. The authors do not report the concentration of the peptides incubated with HRP-9E1 that was later used to detect binding to RBD. In a typical competition ELISA set up, the peptides needs to be titrated when being incubated with 9E1. Blocking of 9E1 binding to RBD needs to be presented as factor of peptide concentration. This will strengthen the message being passed in panels B, C and D in Fig 4.
  7. In Fig 4F and 4G, the authors show the binding region of 9E1 mapped onto spike protein. Could they comment whether the region is accessible in both up and down conformation of RBD or is it limited to only one?
  8. In the discussion the authors mention that the binding epitope of 9E1, while being conserved in different variants of SARS-CoV-2, it is not conserved in SARS-CoV. I would recommend to show this information in the context of panel Fig 4E. It would also be good to compare this sequence with those from the circulating corona viruses. This would highlight how sensitive 9E1 can be as a tool in diagnostics.
  9. This manuscript will benefit from determining the binding strength of 9E1 to RBD, Spike protein or at least to the target peptide. Qualitatively describing either binding in ELISA under serial antibody dilution or measuring affinity in ITC or SPR would significantly improve the report on this antibody. As the authors mention in the discussion, the neutralizing capacity of this antibody or targeting the epitope in this region still remains unknown. If the authors would like to report more on the binding property but not on the functional nature of the antibody, qualitative assessment of binding in comparison to other antibodies would be good to have.

Author Response

  1. In Fig 1C and 1D, the authors show binding of polyclonal antibodies from patient plasma to RBD and Spike protein. Here the negative controls are missing. It is critical to perform the same assay using a plasma sample from non-exposed volunteer or ideally a plasma sample collected from pre-pandemic time point.

Response 1:

Thank you for the comment. We have used three plasma samples from healthy subjects as control. We did not observe nonspecific reaction of control sera to RBD and S protein. The ELISA result was shown in Figure S1, please see it in the supplementary pdf file.

  1. In Fig2A, the FACS panels should be depicted not always as histograms but as dot plot. This is particularly relevant to appreciate the cell frequency and the MFI values in the IgG+ cells stained for RBD-APC.

Response 2:

Thank you for the comments. Please see the new Figure 2A in Page 8.

  1. In Fig 2B, what was the concentration of antibodies tested for binding to RBD? This needs to be mentioned both in the methods and the figure legend. Also, were these experiments performed at serially diluted antibody concentrations as described in the method? If yes, could the author show the dilution curves of all antibodies? It is clear all of them bind to RBD and 9E1 may not be the best antibody. It would be good to compare the binding strength of the antibodies and then show how 9E1 is the only antibody that binds in Western blot.

Response 3:

Thank you for the comments. Firstly, to verify the binding abilities of eight isolated antibodies to RBD, plasmids containing the paired heavy and light chains of antibodies were co-transfected into 293 cells and further evaluated the binding activity to RBD. Due to the tested antibody in supernatant was not purified, we can’t show the exact antibody concentration in Figure 2B. Here, we have performed a new ELISA to confirm the similar binding activity of the purified antibodies at serially diluted concentrations to RBD in Figure 2C (See Page 8).

  1. In Fig 3E, the positive control is missing in IHC.

Response 4:

Thank you for your valuable suggestion. As we mentioned in this paper, few S protein-specific antibodies for IHC assay are available. So, positive control antibody was not shown in IHC.  Instead, to make sure the tissue used in IHC assay was positive to SARS-CoV-2, we used RT-PCR to confirm the tissue is SARS-CoV-2 virus-positive.

  1. In Fig 4. Panel A shows the peptide sequence covered from the protein and whether 9E1 binds to the peptide in Western blot. Here I would recommend the authors to write the amino acid sequences of the peptides used (as shown in panel D) or at least provide them in supplementary figures. In addition, it is important to show the raw data of Western blot as this is a crucial data supporting the message of the manuscript. The authors should show the raw Western blot data of binding to the peptides in comparison to the whole protein.

Response 5:

Thank you for the detailed suggestion. The peptide sequences from panel A to panel C were too long to display in one figure suitably. Instead, we provide the related peptide sequences as Table S1 (see supplementary pdf file).

Figure 4A summarized the activity of 9E1 in western blot. In addition, we have added the SDS-PAGE and western blot (anti-His tag) results of CRMA-fused truncated RBD. We also displayed the western blot result of 9E1 against truncated RBD and the whole RBD protein. Please see the results in Figure S2 (see supplementary pdf file).

  1. Similarly, in Fig 4B, the peptide amino acid sequence should be shown and the competition ELISA values should be shown in comparison to the corresponding controls. The authors do not report the concentration of the peptides incubated with HRP-9E1 that was later used to detect binding to RBD. In a typical competition ELISA set up, the peptides needs to be titrated when being incubated with 9E1. Blocking of 9E1 binding to RBD needs to be presented as factor of peptide concentration. This will strengthen the message being passed in panels B, C and D in Fig 4.

Response 6:

Thank you for your good suggestion. In fact, the competition ELISA was performed with serially diluted 9E1 against peptides at the concentration of 10 μg/mL and we chose part of the ELISA results shown in Figure 4 to explain the key epitope recognized by 9E1. Here, we have added the corresponding detailed ELISA result in Figure S3 (see supplementary pdf file). We are sorry for not providing the result of 9E1 against titrated peptides, while we could find out that the peptides colored in red significantly reduced 9E1 binding to RBD protein at different antibody concentrations.

  1. In Fig 4F and 4G, the authors show the binding region of 9E1 mapped onto spike protein. Could they comment whether the region is accessible in both up and down conformation of RBD or is it limited to only one?

Response 7:

Thank you for your comment. We have proved that 9E1 recognized a linear epitope and any conformational change of spike protein should have no influence on 9E1 binding. Considering that the epitope recognized by 9E1 was exposed at the spike protein surface without coverage by other domains, we hypothesize that the region is accessible in both up and down conformation.

  1. In the discussion the authors mention that the binding epitope of 9E1, while being conserved in different variants of SARS-CoV-2, it is not conserved in SARS-CoV. I would recommend to show this information in the context of panel Fig 4E. It would also be good to compare this sequence with those from the circulating corona viruses. This would highlight how sensitive 9E1 can be as a tool in diagnostics.

Response 8:

The related original paragraph in our paper is ‘The sequence analysis also suggested that amino acid sequence of the 9E1 epitope was highly conserved in more than 1,000 strains of SARS-CoV-2, while the epitope in SARS-CoV-2 differed from corresponding region of S protein from SARS-CoV (data not shown), which indicating 9E1 is possibly a SARS-CoV-2 specific antibody and may not have cross-reactivity to SARS-CoV S protein.’ We want to emphasize that 9E1 epitope is highly conserved in SARS-CoV-2 and 9E1 may be a SARS-CoV-2 specific antibody without cross-reaction with SARS-CoV.

In addition, it is really a great suggestion as Reviewer pointed out that we can compare this sequence with those from other human coronaviruses to highlight the sensitivity of 9E1 as a diagnostic tool. We have added the alignment of S protein sequences of all six human coronaviruses. Part of the alignment is shown below and we find out that the region (marked in red) is variable (see Figure S5. in supplementary pdf file). The sequence recognized by 9E1 is totally different from those of other human coronaviruses which further confirms the sensitivity of 9E1.

  1. This manuscript will benefit from determining the binding strength of 9E1 to RBD, Spike protein or at least to the target peptide. Qualitatively describing either binding in ELISA under serial antibody dilution or measuring affinity in ITC or SPR would significantly improve the report on this antibody. As the authors mention in the discussion, the neutralizing capacity of this antibody or targeting the epitope in this region still remains unknown. If the authors would like to report more on the binding property but not on the functional nature of the antibody, qualitative assessment of binding in comparison to other antibodies would be good to have.

Response 9:

Thank you for the comments. We have added qualitatively assessment of 9E1 binding in ELISA as shown in updated Figure 3C and 3D (See the revision in Page 10). We report 9E1 mostly in order to introduce an antibody recognizing a novel linear epitope with potential application as a diagnostic tool. In fact, there are few reported antibodies like 9E1. We are very willing to make such comparison while it needs more time to search and prepare reported antibodies.

Round 2

Reviewer 2 Report

Dear authors,

The revised version of the manuscript and the response comments addressed all the questions raised by me. I have a few minor comments to the authors mentioned below. With this, I support the manuscript for publishing.

  1. In response 4, the authors mentioned the verification of infected tissue used in IHC by RT-PCR. I would recommend to add this to the methods.
  2. Similarly, the response point 7 could be included in the discussion.
  3. The citation in line 264, 266 and 268-269 should be Fig. S3A, S3B and S3C, respectively.

I would recommend the authors to closely check the citations to the revised figure panels.

Revised title misses the "linear epitope" feature of the antibody. Given the response authors make in point 9 and the abstract, the title will benefit from mentioning linear epitope. Also, I'm not sure "monospecific" is the correct phrase. I would recommend "Rabbit monoclonal antibody specifically recognizing a linear epitope in the RBD of SARS-CoV-2 spike protein".

Author Response

Point-by-point responses to reviewer 2 (Round 2)

1. In response 4, the authors mentioned the verification of infected tissue used in IHC by RT-PCR. I would recommend to add this to the methods.

Response 1:

Thank you for your suggestion. We have added the description of RT-PCR to the methods in Lines 158-164, Page 4.

2. Similarly, the response point 7 could be included in the discussion.

Response 2:

Thank you. We have added the sentences below to discussion in Lines 336-338, Page 13.

3. The citation in line 264, 266 and 268-269 should be Fig. S3A, S3B and S3C, respectively.

Response 3: Thank you. We have revised them, please see Pages 11-12, Lines 276, 278, 280-281 in the revision.

4. I would recommend the authors to closely check the citations to the revised figure panels.

Response 4: Thank you for your suggestion. We have checked all citations to the revised figure panels carefully again. No other mistake is found now.

5. Revised title misses the "linear epitope" feature of the antibody. Given the response authors make in point 9 and the abstract, the title will benefit from mentioning linear epitope. Also, I'm not sure "monospecific" is the correct phrase. I would recommend "Rabbit monoclonal antibody specifically recognizing a linear epitope in the RBD of SARS-CoV-2 spike protein".

Response 5: Thanks for your great suggestion. We agree to change the title to “Rabbit monoclonal antibody specifically recognizing a linear epitope in the RBD of SARS-CoV-2 spike protein” as reviewer suggested. Please see the new title in the revision.